# Efficient and Light-Weight Federated Learning via Asynchronous Distributed Dropout

**Chen Dun**
Rice University

**Mirian Hipolito**
Microsoft Research

**Chris Jermaine**
Rice University

**Dimitrios Dimitriadis**
Microsoft Research

**Anastasios Kyrillidis**
Rice University

## Abstract

We focus on dropout techniques for asynchronous distributed computations in federated learning (FL) scenarios. We propose `AsyncDrop`, a novel asynchronous FL framework with smart (i.e., informed/structured) dropout that achieves better performance compared to state of the art asynchronous methodologies, while resulting in less communication and training time costs. The key idea revolves around sub-models out of the global model, that take into account the device heterogeneity. We conjecture that such an approach can be theoretically justified. We implement our approach and compare it against other asynchronous baseline methods, by adapting current synchronous FL algorithms to asynchronous scenarios. Empirically, `AsyncDrop` significantly reduces the communication cost and training time, while improving the final test accuracy in non-i.i.d. scenarios.

## 1 Introduction

**Background on Federated Learning (FL).** Federated Learning (FL) [21, 17, 15] is a distributed learning protocol that has witnessed fast development the past demi-decade, as it deviates from the traditional distributed learning paradigms and allows the integration of small computing (edge) devices —such as smartphones [30], drones [25], and IoT devices [22]— in the learning procedure. Such real-life, edge devices though are extremely *heterogeneous* [33]: they have drastically different specifications in terms of compute power, device memory and communication bandwidths. Directly applying common *synchronized* FL algorithms –such as FedAvg and FedProx [17, 21] that require full model broadcasting and global synchronization– results often in a "stragglers" effect [23, 13, 31].

**The dominant use of synchronous computations.** One way to handle such issues is by utilizing *asynchrony* instead of *synchrony* in the learning process. To explain the main differences, let us first set up the background. In a synchronous distributed algorithm, a global model is usually stored at a central server and is broadcast periodically to all the participating devices. Then, each device performs some local training steps on its own model copy, before the device sends the updated model to the central server. Finally, the central server updates the global model by aggregating the received updated model copies. This protocol is followed in most FL protocols, including the most well-known baselines like FedAvg [21], FedProx [17], FedNova [34] and SCAFFOLD [15]. The main criticism against synchronous learning could be that it often results in heavy communication-/computation-overheads when mobile devices are used, long idle/waiting times for workers and stragglers effects.

**Main dispute in asynchronous learning.** While asynchrony in computations seems a favorable alternative, the development of a asynchronous learning method, however, is often convoluted. In the past decade, HogWild! [24, 20] has emerged as a general asynchronous distributed methodology, and has been applied initially in basic ML problems like sparse linear/logistic regression [42, 41, 12]. Ideally, based on sparsity arguments, each edge device can independently update parts of the global model –that overlap only slightly with the updates of other workers– in a lock-free fashion [24, 20]. This way, faster, more powerful edge workers do not suffer from idle waiting due to slower stragglers.

Workshop on Federated Learning: Recent Advances and New Challenges, in Conjunction with NeurIPS 2022 (FL-NeurIPS'22). This workshop does not have official proceedings and this paper is non-archival.

Yet, the use of asynchrony has been a topic of dispute in distributed neural network training [5, 2]. Asynchronous training often inherently suffers from lower accuracy as compared to synchronized SGD, which results in the domination of synchronized SGD in neural network training [2].

**Resurgence in asynchrony.** Recently, asynchronous methods have regained popularity, mainly due to the interest in applying asynchronous motions within FL on edge devices. Yet, traditional off-the-self asynchronous algorithms still have issues, which might be exacerbated in the FL setting. As slower devices take longer local training time before updating the global model, this might result in inconsistent update schedule of the global model, compared to that of faster devices. This might have negative ramifications: $i$) For FL on i.i.d. data, this will cause the gradient staleness problem and result in convergence rate decrease; and, $ii$) on non-i.i.d. data, this will result in significant drop in global model final accuracy.

To overcome such problems, fresh proposals on asynchronous FL develop weighted global aggregation techniques that take into consideration the heterogeneity of the devices [38, 3, 28]; yet, these methods often place a heavy computation/communication burden, as they rely on broadcasting the whole model's updates to all the clients and/or the server. Other works monitor client speed to help training assignment [18, 1]. Finally, recent efforts propose semi-asynchronous methods, where participating devices are selected and buffered in order to complete a semi-synchronous global update periodically [13, 37]. A thorough discussion on asynchronous methods in FL can be found in [39].

**What is different in this work?** As most algorithms stem from directly adapting asynchrony in synchronous FL, *one still needs to broadcast full model to all devices, following a data parallel distributed protocol [9, 26], regardless of device heterogeneity*. This inspire us to ask a key question:

> *"Can we select sub-models out of the global model and send these to each device, taking into account the device heterogeneity? Potentially, this could further reduce the negative effect of gradient staleness and overfitting problem in asynchronous FL."*

We answer this question affirmatively, by proposing a novel distributed dropout method for FL. We dub our method `AsyncDrop`. Our approach assigns different sub-models to each device[1]; empirically, such a strategy decreases the required time to converge to an accuracy level, while preserving favorable final accuracy. This work reinstitutes the dispute between synchrony and asynchrony and focuses on asynchronous computations should in heterogeneous distributed scenarios, as in FL. Our idea is based on the ideas of HogWild! [24, 20] –in terms of sparse sub-models– and Independent Subnetwork Training (IST) [40, 8, 19, 36] –where sub-models are deliberately created for distribution, in order to decrease both computational and communication requirements.

Yet, we deviate from these works in the following sense: $i$) To the best of our knowledge, we are not aware of any fully asynchronous FL implementations; the combination of HogWild! and IST ideas was not stated and tested before this work. $ii$) While HogWild!-line of work provides optimization guarantees, we are the first to consider the non-trivial, non-convex neural network setting and provide theoretical guarantees for convergence; such a result combines tools from asynchronous optimization, Neural Tangent Kernel assumptions and focuses on convolutional neural networks, that deviate from fully-connected layer simplified scenarios. Finally, $iii$) we provide system-level algorithmic solutions for our approach, mirroring best-practices found during our experiments. Overall, the contributions of this work can be summarized as follows:

- We consider and propose *asynchronous distributed dropout* for efficient large-scale FL. Our focus is on non-trivial, non-convex ML models –as in neural network training– and our framework provides specific engineering solutions how to handle these cases in practice.
- We theoretically characterize and support our proposal with rigorous and non-trivial convergence rate guarantees. Currently, our conjecture assumes bounded delays; our future goal is to exploit recent developments on asynchronous optimization theory that drop such assumptions. Yet, our theory considers the harder case of neural network training, which is often omitted in existing –but rather abstract– theory results.
- We provide specific implementation instances and share "best practices" for faster distributed FL neural network training in practice. Our preliminary results include baseline asynchronous implementations of many synchronous methods (such as FedAvg, FedProx, and more), that are not existent currently, to the best of our knowledge.

---

[1] We consider both random assignment, as well as structured assignments, based on the computation power of the devices.

## 2 Problem Setup and Challenges

**Federated Learning formulation.** We use the following notation for data representation: given a data source distribution $\mathcal{D}$, $\{\mathbf{x}_i, y_i\} \sim \mathcal{D}$ represents a data point where $\mathbf{x}_i$ is a data sample, and $y_i$ is its corresponding label (for classification purposes). We use $n$ to denote the total number of samples, unless otherwise stated. Let $S$ be the total number of clients in a distributed FL scenario. Each client $i$ has its own local data $\mathcal{D}_i$ such that the whole dataset satisfies $\mathcal{D} = \cup_i \mathcal{D}_i$, and usually $\mathcal{D}_i \cap \mathcal{D}_j = \emptyset, \forall i \neq j$. The goal of federated learning is to find a global model $\mathbf{W}$ that achieves good accuracy on all data $\mathcal{D}$, by minimizing the following optimization problem:

$$\mathbf{W}^\star = \underset{\mathbf{W} \in \mathcal{H}}{\operatorname{argmin}} \left\{ \mathcal{L}(\mathbf{W}) := \tfrac{1}{S} \sum_{i=1}^{S} \ell(\mathbf{W}, \mathcal{D}_i) \right\}, \text{ where } \ell(\mathbf{W}, \mathcal{D}_i) = \tfrac{1}{|\mathcal{D}_i|} \sum_{\{\mathbf{x}_j, y_j\} \in \mathcal{D}_i} \ell(\mathbf{W}, \{\mathbf{x}_j, y_j\})$$

With a slight abuse of notation, $\ell(\mathbf{W}, \mathcal{D}_i)$ denotes the *local* loss function for user $i$, associated with a local model $\mathbf{W}_i$ (not indicated above), that gets aggregated with the models of other users. Local data distribution $\mathcal{D}_i$ can be heterogeneous and follow a non-i.i.d. distribution. In the following sections, we consider both i.i.d. case and non-i.i.d. case.

**Details of asynchronous training.** An abstract description of how asynchronous FL operates is provided in Algorithm 1. In particular, given a number of server iterations $T$, each client $i$ gets the updated global model $\mathbf{W}_t$ from the server, and further locally trains it using $\mathcal{D}_i$ for a number of local iterations $l$.[2] Asynchronous federated learning assumes each client has different computation power and communication bandwidth; this can be abstracted by different wall-clock times required to finish a number of local training iterations. Thus, when client $i$ has completed its round, the updated model is shared with the server to be aggregated, before the next round of communication and computation starts for client $i$. This is different from classical synchronous federated learning, where the global model is updated only when all participating clients finish certain local training iterations.

---

**Algorithm 1** Meta Asynchronous FL

**Parameters**: $T$ iters, $S$ clients, $l$ local iters., $\mathbf{W}$ as current global model, $\mathbf{W}_i$ as local model for $i$-th client, $\alpha \in (0, 1)$, $\eta$ step size.

─────────── $\infty$ ───────────

$\mathbf{W} \leftarrow$ randomly initialized global model.
`//On each client` $i$ `asynchronously:`
**for** $t = 0, \ldots, T - 1$ **do**
  $\mathbf{W}_{i,t} \leftarrow \mathbf{W}$
  `//Train` $\mathbf{W}_i$ `for` $l$ `iters. via SGD`
  **for** $j = 1, \ldots, l$ **do**
    $\mathbf{W}_{i,t} \leftarrow \mathbf{W}_{i,t} - \eta \frac{\partial \mathcal{L}}{\overline{\mathbf{W}_{i,t}}}$
  **end for**
  `//Write local to global model`
  $\mathbf{W} \leftarrow (1 - \alpha) \cdot \mathbf{W} + \alpha \cdot \mathbf{W}_{i,t}$
**end for**

---

## 3 Asynchronous Distributed Dropout in the i.i.d. case

**Intricacies of asynchronous FL.** Asynchronous motions often lead to inconsistent update schedules of the global model and are characterized by gradient staleness and overfitting. While there are some recent solutions [38, 3, 28, 18, 1, 13, 37], real-life FL applications include edge devices with limited communication and computation capabilities that further exacerbate such issues(e.g., how often and fast they can connect with the central server, and how powerful as devices they are). For instance, edge devices such as IoT devices or mobile phones [22], might only be able to communicate with the server within short time windows, due to network conditions or user permission policy (e.g., a device might need to upload and download the updated model in a short time window). Such edge devices have limited computational capabilities, as compared to GPU-based systems.

Finally, existing semi-asynchronous methods [38, 3, 28, 18, 1, 13, 37] might require fast clients wait until all other clients' updates are completed, in order to receive the updated model for the next round of local training. This might further result in nondeterministic long waiting time.

**(Distributed) Dropout**. Dropout [32, 29, 10, 4] is a widely-accepted regularization technique in deep learning. The procedure of Dropout is as follows: per training round, a random mask over the parameters is generated; this mask is used to nullify part of the neurons in the neural network for this particular iteration. Variants of dropout include the drop-connect [32], multisample dropout [14], Gaussian dropout [35], and the variational dropout [16].

The idea of dropout has also been used in efficient distributed and/or federated learning scenarios. [11] introduces FjORD and the Ordered Dropout, a *synchronous* distributed dropout technique that

---
[2]Details on the use of the optimizer, how it is tuned with respect to step size, mini-batch size, etc. are intentionally hidden at this point of the discussion.

leads to ordered, nested representation of knowledge in models, and enables the extraction of lower footprint submodels without the need of retraining. Such submodels are more suitable in client system heterogeneity, as they adapt submodel's width to the client's capabilities. Similar approaches include Nested Dropout [27] and HeteroFL [6].[3]

**Our proposal and main hypothesis.** In this work, we focus on the *asynchronous version of distributed dropout*. We first study theoretically whether asynchrony provably works in non-trivial non-convex scenarios –as in training convolutional neural networks– with random masks that generate submodels for each worker/participating device. The algorithm we focus on is described in Algorithm 2 and is based on recent distributed protocols [40, 8, 19, 36]. The main difference of Algorithm 2, compared to Algorithm 1, is that the former splits the model vertically per iteration, where each submodel contains all layers of the neural network, but only with a (non-overlapping) subset of neurons being active in each layer. Multiple local SGD steps can be performed without the workers having to communicate.

---

**Algorithm 2** `AsyncDrop` algorithm

---

**Parameters**: $T$ iters, $S$ clients, $l$ local iters., $\mathbf{W}$ as current global model, $\mathbf{W}_i$ as local model for $i$-th client, $\alpha \in (0,1)$, $\eta$ step size.

────────── ∞ ──────────

$\mathbf{W} \leftarrow$ randomly initialized global model.
`//On each client i asynchronously:`
**for** $t = 0, \ldots, T-1$ **do**
   Generate mask $\mathbf{M}_{i,t}$
   $\mathbf{W}_{i,t} \leftarrow \mathbf{W}_t \odot \mathbf{M}_{i,t}$
   `//Train` $\mathbf{W}_{i,t}$ `for` $l$ `iters.  via SGD`
   **for** $j = 1, \ldots, l$ **do**
      $\mathbf{W}_{i,t} \leftarrow \mathbf{W}_{i,t} - \eta \frac{\partial \mathcal{L}}{\overline{\mathbf{W}_{i,t}}}$
   **end for**
   `//Write local to global model`
   $\mathbf{W}_{t+1} \leftarrow \mathbf{W}_t \odot (\mathbf{M}_{i,t})^c$
            $+((1-\alpha)\cdot\mathbf{W}_t + \alpha\cdot\mathbf{W}_{i,t}) \odot \mathbf{M}_{i,t}$
**end for**

---

**Theoretical Results**. We perform theoretical analysis on a one-hidden-layer convolutional neural network (CNN), and show the convergence with random filter dropout. Consider a training dataset $(\mathbf{X}, \mathbf{y}) = \{(\mathbf{x}_i, y_i)\}_{i=1}^n$, where each $\mathbf{x}_i \in \mathbb{R}^{\hat{d} \times p}$ is an image and $y_i$ being its label. Here, $\hat{d}$ is the number of input channels and $p$ the number of pixels. Let $q$ denote the size of the filter, and let $m$ be the number of filters in the first layer. We let $\hat{\phi}(\cdot)$ denote the patching operator with $\hat{\phi}(x) \in \mathbb{R}^{q\hat{d} \times p}$ [7]. Consider the first layer weight $\mathbf{W} \in \mathbb{R}^{m \times q\hat{d}}$, and second layer (aggregation) weight $\mathbf{a} \in \mathbb{R}^{m \times p}$. We assume that only the first layer weights $\mathbf{W}$ is trainable. The CNN model, trained on the means squared error, has the form:

$$f(\mathbf{x}, \boldsymbol{\zeta}) = \left\langle \mathbf{a}, \sigma\left(\mathbf{W}\hat{\phi}(\mathbf{x})\right)\right\rangle; \quad \mathcal{L}(\mathbf{W}) = \|f(\mathbf{X}, \mathbf{W}) - \mathbf{y}\|_2^2,$$

where $\boldsymbol{\zeta}$ abstractly represents all training parameters, $f(\mathbf{x}, \cdot)$ denotes the output of the one-layer CNN for input $\mathbf{x}$, and $\mathcal{L}(\cdot)$ is the loss function.

**Assumption 3.1** (*Training Data*) *Assume that for all data points, we have* $\|\mathbf{x}_i\|_F = q^{-\frac{1}{2}}$ *and* $|y_i| \leq C$ *for some constant* $C$. *Moreover, for all* $i, i'$ *we have* $\mathbf{x}_i \neq \mathbf{x}_{i'}$.

This assumption can be easily satisfied with data normalization. For simplicity, let $d := q\hat{d}$.

**Assumption 3.2** (*Initialization*) $\mathbf{w}_{0,r} \sim \mathcal{N}\left(0, \kappa^2 \mathbf{I}\right)$ *and* $a_{rj} \sim \left\{\pm\frac{1}{p\sqrt{m}}\right\}$ *for* $r \in [m]$ *and* $j \in [p]$.

In an asynchronous scenario, we assume the neural network weights are updated with stale gradients: here, $d_t$ denotes the delay at training step $t$. Further, for all $t$ iterations, we assume $d_t$ is bounded by a constant $E$. Then, gradient descent motions with delays take the form:

$$\mathbf{W}_t = \mathbf{W}_t - \eta \nabla_{\mathbf{W}} \mathcal{L}\left(\mathbf{W}_{t-d_t}\right), \quad \text{where } d_t \leq E.$$

Based on the above we make the following conjecture:

---

[3]Beyond the restriction to synchronous training, such approaches often lack a specific instantiation of the proposed methodology to neural network training. This present work proposes specific algorithmic solutions on how to consolidate these ideas into a single practical framework for neural network training in FL scenarios.

**Theorem 3.1** *Let $f(\cdot, \cdot)$ be a one-hidden-layer CNN with the second layer weight fixed. Let $\mathbf{u}_t$ abstractly represent the output of the model after $t$ iterations, over the random selection of the masks. Let $\xi$ denote the dropout rate ($\xi = 1$ dictates that all neurons are selected), and denote $\theta = 1 - (1-\xi)^S$ the probability that a neuron is active in at least one subnetwork. Assume the number of hidden neurons satisfies $m = \Omega\left(\max\{\frac{n^4 K^2}{\lambda_0^4 \delta^2}\max\{n,d\}, \frac{n}{\lambda_0}\}\right)$ and the step size satisfies $\eta = O\left(\frac{\lambda_0}{n^2}\right)$. Let $\kappa$ be a proper initialization scaling factor, and it is considered constant. We use $\lambda_0$ to denote the smallest eigenvalue of the Neural Tangent Kernel matrix. Let Assumptions 1 and 2 be satisfied. Then, the following convergence rate guarantee is proved to be satisfied:*

$$\mathbb{E}_{\mathbf{M}_t}\left[\|\mathbf{u}_{t+1} - \mathbf{y}\|_2^2\right] \leq \left(1 - \frac{\theta\eta\lambda_0}{4}\right)^t \|\mathbf{u}_0 - \mathbf{y}\|_2^2$$
$$+ O\left(\frac{\theta\eta\lambda_0^3\xi^2\kappa^2 E^2}{n^2} + \frac{\xi^2(1-\xi)^2\theta\eta n^3\kappa^2 d}{m\lambda_0} + \frac{\eta^2\theta^2 n\kappa^2\lambda_0\xi^4 E^2}{m^4} + \frac{\xi^2(1-\xi)^2\theta^2\eta^2 n^2\kappa^2 d}{m^3\lambda_0}\right.$$
$$\left. + \frac{\xi^2(1-\xi)^2\theta^2\eta^2\kappa^2\lambda_0 E^2}{m^3} + \frac{\xi^2(1-\xi)^2\theta^2\eta^2 n^2\kappa^2 d}{m^2\lambda_0} + \frac{n\kappa^2\left(\theta - \xi^2\right)}{S}\right)$$

## 4 Smart Partition in Asynchronous Dropout for FL

**Potential issues in asynchronous FL.** Despite theoretical support, asynchronous algorithms might show slower convergence rates and/or lower final model accuracies, mainly due to inconsistent global model updates between slower devices and faster devices. With non-i.i.d. data as in the case of FL, more frequent updates by faster devices could lead to global model overfitting over local data on the same devices; this often results in significantly sub-optimal final model accuracy. *These facts suggest a more careful handling of model splitting and model distribution among heterogeneous workers.*

---

**Algorithm 3** `Smart Dropout` for Asynchronous FL

---

**Parameters**: $T$ iters, $S$ clients, $l$ local iters., $\mathbf{W}$ as current global model, $\mathbf{W}_i$ as local model for $i$-th client, $\eta_g$ as global LR, $q(\cdot)$ weight score function, $\psi(i)$ computes the computation capacity of $i$-th worker, $\varphi(\mathbf{W}, \psi(i), q(\cdot))$ is the `Smart Dropout` function that creates the mask, based on worker capacity $\psi$ and score function $q$, $\alpha \in (0, 1)$.

---∞---

$\mathbf{W} \leftarrow$ randomly initialized global model.
//On each client $i$ asynchronously:
**for** $t = 0, \ldots, T-1$ **do**
  //For $i$-th fastest worker, $\varphi(\cdot)$ drops
  weights with $i$-th largest $q(\cdot)$ score
  Generate mask $\mathbf{M}_{i,t} = \varphi(\mathbf{W}_t, \psi(i), q(\cdot))$
  $\mathbf{W}_{i,t} \leftarrow \mathbf{W}_t \odot \mathbf{M}_{i,t}$
  //Train $\mathbf{W}_{i,t}$ for $l$ iters. via SGD
  **for** $j = 1, \ldots, l$ **do**
    $\mathbf{W}_{i,t} \leftarrow \mathbf{W}_{i,t} - \eta_g \frac{\partial \mathcal{L}}{\mathbf{W}_{i,t}}$
  **end for**
  **if** $i$-th client is fastest **then**
    Update $\eta_g$
  **end if**
  //Write local to global model
  $\mathbf{W}_{t+1} \leftarrow \mathbf{W}_t \odot (\mathbf{M}_{i,t})^c$
    $+ ((1-\alpha)\cdot\mathbf{W}_t + \alpha\cdot\mathbf{W}_{i,t}) \odot \mathbf{M}_{i,t}$

  //Update score $q$
  $q(\mathbf{W}_{t+1}^j) = \left\|\mathbf{W}_{t+1}^j - \mathbf{W}_0^j\right\|_1, \forall j \in \mathcal{J}$
**end for**

---

`Smart Dropout`: **improved asynchronous solution**. We propose to resolve inconsistency in global model updates by "balancing" the contribution from different devices with a `Smart Dropout` method. Briefly, `Smart Dropout` assigns different weights to different devices, based on the updated rate of the weights, and the computation power of the devices.

An algorithmic description is provided in Algorithm 3. We assign model weights that are less updated to the faster devices; this way, we guarantees that such weights will be updated more frequently than random weight assignment. Similarly, we assign model weights that are updated more often – and, thus, converging faster– to the slower devices. The premise behind such a protocol is that *all weights, eventually, will be updated/will converge with a similar rate.*

In the i.i.d. case, this would reduce the gradient staleness: slower devices take fewer steps to converge on the assigned weights; this is compensated by assigning "fast-converging weights", that were assigned to faster devices in the previous iterations. In the non-i.i.d. case, such a strategy would reduce any overfitting over the local data in faster devices: The weights previously updated the most by the faster devices will not be assigned to the fast devices in the near-future iterations.

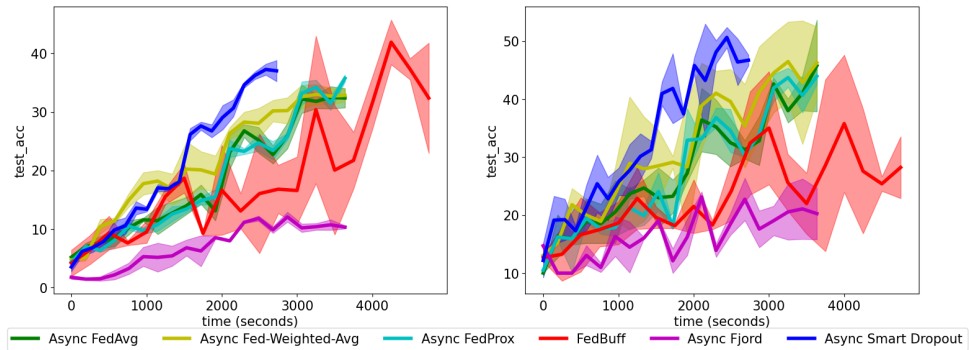

Figure 1: *Left:* CIFAR10 non-i.i.d.; ***Right:*** CIFAR100, non-i.i.d..

`Smart Dropout`**: its ingredients**. The above are encapsulated with a weight score function $q(\cdot)$, a device-capacity score function $\psi(\cdot)$ and the mask generator function $\varphi(\cdot)$ in Algorithm 3. The function $q(\cdot)$ quantifies the update speed of each weight. One simple score can be the norm of weight change: Given predefined grouping of weights in set $\mathcal{J}$, define $\mathbf{W}_t^j$ for $j \in \mathcal{J}$ as the $j$-th weight/group of weights of global network at the $t$-th update. Then, the score function is selected as: $q(\mathbf{W}_t^j) = \left\| \mathbf{W}_t^j - \mathbf{W}_0^j \right\|_1$, i.e., the score function measures *how far from the initial values the $j$-th group of weights has moved*. In our experiments, we have tested grouping of weights $\mathcal{J}$ by filters and by layers. Further, we have some measure of the computation capacity of each device $\psi(i)$, and order the devices in descending order such that the first device is the fastest and last the slowest.

The mask generator function $\varphi(\mathbf{W}, \psi(i), q(\cdot))$ generates masks that nullify weights in $\mathbf{W}$, based on the score function $q$ over the weights, and the device capacity list in $\psi(\cdot)$. The `Smart Dropout` strategy is that for the $i$-th fastest worker, we drop weights with $i$-th largest $q$ score.

## 5 Experiments

We focus on *fully asynchronous FL using CNNs.* Yet, we highlight that, with proper tuning, our ideas are model agnostic by design. To simulate the heterogeneous edge devices, we implement devices as different processes, while we manually insert different time delays to simulate different compute nodes and communication bandwidths. *In these preliminary results, we focus only on non-i.i.d. scenarios.* We consider the difficult case where data distribution will be correlated to computation power, which results in a more heterogeneous model update during training. In all experiments, we use 104 devices/users in total and activate 8 of them at any moment. In all `Smart Dropout` experiments, we use 25% dropout rate.

**Baselines**. We compare asynchronous implementations of five algorithms: $i$) asynchronous FedAvg; $ii$) asynchronous FedAvg with weighted aggregation based on gradient staleness; $iii$) asynchronous FjORD; $iv$) asynchronous FedProx; and $iv$) FedBuff. These baselines are straightforward implementations of synchronous versions (except from FedBuff).

**Preliminary results**. We only highlight key points in our preliminary results; more findings and further discussion will be included during the workshop. Focusing on the non-i.i.d. case for CIFAR10 (Table 1), we observe that our proposal gets the maximum accuracy compared to the rest of the algorithms, while being stable: notice the high variability in values coming from different runs of our code (indicated with **red colored text**). At the same time, our algorithm is faster and requires less communication over the course of the execution.

Similar observations can be made for the case of non-i.i.d. CIFAR100 (Table 2). Here, observe that FedBuff achieves the best maximum accuracy. However, again the achieved models are not consistent over various runs of the algorithm; our algorithm shows a stable performance, within reach of the high variance by FedBuff. At the same time, our algorithms shows great performance in terms of acceleration in execution and required communication bandwidth. This behavior is further highlighted in Figure 1.

Table 1: Test accuracy on non-i.i.d. CIFAR10 datasets.

| | Max. Accuracy | Time to 35% accuracy | Time Saved Ratio | Comm. Saved Ratio |
|---|---|---|---|---|
| AsyncDrop | **50.67** $\pm$ 1.75 | **1579.1s** | - | |
| Async FedAvg | 45.79 $\pm$ **7.9** | 2105.5s | -33.34% | -15.56% |
| Async Fed-Weighted-Avg | 46.51 $\pm$ **6.8** | 2105.5s | -33.34% | -15.56% |
| Async FedProx | 43.97 $\pm$ 1.35 | 2296.9s | -45.46% | -26.06% |
| Async Fjord | 23.14 $\pm$ 0.90 | N/A | N/A | N/A |
| FedBuff | 35.81 $\pm$ **11.83** | 2296.9s | -45.46% | -443.03% |

Table 2: Test accuracy on non-i.i.d. CIFAR100 datasets.

| | Max. Accuracy | Time to 32% accuracy | Time Saved Ratio | Comm. Saved Ratio |
|---|---|---|---|---|
| AsyncDrop | 37.26 $\pm$ 0.93 | **2296.8s** | - | |
| Async FedAvg | 32.47 $\pm$ 1.89 | 3062.5s | -33.38% | -15.56% |
| Async Fed-Weighted-Avg | 32.98 $\pm$ 1.71 | 3062.5s | -33.38% | -15.56% |
| Async FedProx | 35.75 $\pm$ 0.61 | 3062.5s | -33.38% | -15.56% |
| Async Fjord | 12.07 $\pm$ 0.83 | N/A | N/A | N/A |
| FedBuff | **41.91** $\pm$ **3.80** | 3253.9s | -41.72% | -488.28% |

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
