# OpenReview forum: "Efficient and Light-Weight Federated Learning via Asynchronous Distributed Dropout"
_NeurIPS.cc/2022/Workshop/Federated_Learning — FL-NeurIPS 2022 Poster_

### Official Review · Reviewer_vFw2 · 2022-10-15
**an asynchronous FL algorithm with partial model update**

Recently, some Hetero-FL and Fjord (and other followup papers) have proposed to federated learning algorithms allowing each client to only train a submodel (e.g., a subset of neural network weights). The principal motivation of these algorithms is to address system heterogeneity as the size of the submodel can be adapted to client capabilities.
This paper proposes two enhancements to these algorithms:
1) asynchronous operation;
2) a submodel selection strategy which takes into account not only clients' capabilities but also which parameters have been updated the least until now.
It presents a conjectured convergence rate of the proposed asynchronous algorithm (but without taking into account the submodel extraction) for a one-hidden-layer convolutional neural network.

I have found the presentation misleading in many points:
- I have found surprising that Hetero-FL and Fjord are only mentioned at page 4 and not directly in the introduction when the idea of working with submodels is presented as the main contribution of the paper
- In the introduction, the strong focus on the synchrony vs asynchrony "dispute" and the statement "this work reinstitutes the dispute" also leads to suppose that the paper will contribute to such dispute, but this is not the case. While the proposal is supposed to mitigate the asynchrony's problems, there is no comparison with synchronous algorithms in the experimental section
- The introduction also stresses that the paper provides  "theoretical guarantees for convergence" for "non-trivial, non-convex neural network setting", and the result "combines tools for asynchronous optimization, neural tangent kernel assumptions,..." but there are not such guarantees in the paper! There is only a conjecture with no explanation about why should be true or what could be the steps to prove it.
Overall, it seems the introduction describes a different paper, perhaps the paper the authors aimed---but did not succeed---to write.

I find the ideas of limited novelty. The submodel selection part is not very clear and the description is partially incomplete (e.g., how is \eta_g updated in Algorithm 3?). I have also doubts about the fact of focusing training on submodels whose (l1) distance from their initial random configuration is the smallest. Different submodels may converge at different speed even in centralized settings and one could invest more computing resources on submodels which already converged.

Experiments are very preliminary ones: results appear very noisy and curves suggest that training did not finish.

Other comments
- the left and right part of figure 1 have probably been inverted. If that's not the case, the plots do not seem to match results in Table 1.
- the writing is often too colloquial, see expressions like "the main criticism... could be that" "asynchronous algorithms still have issues", "fresh proposals"
- \mathcal D is used to denote both the training dataset and the distribution samples have been drawn from
- HeteroFL appeared on Arxiv at least 4 months earlier than Fjord and was also accepted at an earlier conference (ICLR 2021) than Fjord (NeurIPS 2021). I have found surprising that Fjord's paper does not even mention HeteroFL which had already been presented at ICLR 2021 at the moment of Fjord's submission to  NeurIPS. I think we should give more credits to HeteroFL for the submodel training.

---

### Official Review · Reviewer_z72K · 2022-10-16
**This paper proposes an efficient federated learning method via distributed dropout. The paper addresses a relevant topic. The case studies are well presented. The description of the proposed method seems well. Besides, detailed contributions in the paper are well described.**

This paper proposes an efficient federated learning method via distributed dropout. The paper addresses a relevant topic. The case studies are well presented. The description of the proposed method seems well. Besides, detailed contributions in the paper are well described.
1.	Some mathematical symbols do not explain the meanings (i.e., M), and formulas do not introduce principles. Please use the prescriptive grammar and formula to give the explanation.
2.	How to guarantee and quantity, robustness and convergence performance of the proposed method?
3.	Also, the presented theory is enough, though there are some English inconsistencies that need to be corrected. Careful proofreading is necessary.
4.	There are some hyperparameters and how do you select these? Please show the detailed information about these.

---

### Official Review · Reviewer_Rnbi · 2022-10-19
**Interesting approach, potential for insightful discussions**

Asynchronous FL is an important aspect for cross-device FL settings. While many real-world systems handle it with semi-async approaches or device lock-in/attention mechanisms, there is appetite for novel asynchronous approaches.

This work explores the case of dropouts based "model parallelism" within FL where non-overlapping masks are generated to split the model across clients during each training round. The authors explore the case of random split as well as a score based split to account for device's computation capacity for the split. Initial experiments show promising results.

I only read through the paper once so will give the benefit of doubt to the authors but here are some remarks
- The asynchronous aspect isn't immediately obvious in the notation
- I think the design of mask M_{i,t} is key to AsyncDrop so it would be worth detailing its formalism
- 104 clients with quorum size of 8 seem a bit arbitrary, any reason for this particular choice?
- Was wondering why the termination time for algorithms is different in Fig 1?
- I think the sensitivity of experiments to dropout rate would be critical for assessing the viability of the algo in practice
- It might be useful to refer to server-side iterations as training rounds to avoid the confusion with client-side iterations

---

### Decision · Program_Chairs · 2022-10-20

Accept (Poster)